# Aggregative cycles evolve as a solution to conflicts in social investment

**Leonardo Miele** [1,2]*, **Silvia De Monte** [2,3]*

**1** School of Mathematics, University of Leeds, U.K., **2** Institut de Biologie de l'Ecole Normale Supérieure, Département de Biologie, Ecole Normale Supérieure, CNRS, INSERM, PSL Research University, Paris, France, **3** Department of Evolutionary Theory, Max Planck Institute for Evolutionary Biology, Plőn, Germany

\* mmlm@leeds.ac.uk (LM); silvia.de.monte@bio.ens.psl.eu (SDM)

## Abstract

Multicellular organization is particularly vulnerable to conflicts between different cell types when the body forms from initially isolated cells, as in aggregative multicellular microbes. Like other functions of the multicellular phase, coordinated collective movement can be undermined by conflicts between cells that spend energy in fuelling motion and 'cheaters' that get carried along. The evolutionary stability of collective behaviours against such conflicts is typically addressed in populations that undergo extrinsically imposed phases of aggregation and dispersal. Here, via a shift in perspective, we propose that aggregative multicellular cycles may have emerged as a way to temporally compartmentalize social conflicts. Through an eco-evolutionary mathematical model that accounts for individual and collective strategies of resource acquisition, we address regimes where different motility types coexist. Particularly interesting is the oscillatory regime that, similarly to life cycles of aggregative multicellular organisms, alternates on the timescale of several cell generations phases of prevalent solitary living and starvation-triggered aggregation. Crucially, such self-organized oscillations emerge as a result of evolution of cell traits associated to conflict escalation within multicellular aggregates.

## Author summary

In aggregative multicellular life cycles, cells come together in heterogenous aggregates, whose collective function benefits all the constituent cells. Current explanations for the evolutionary stability of such organization presume that alternating phases of aggregation and dispersal are already in place. Here we propose that, instead of being externally driven, the temporal arrangement of aggregative life cycles may emerge from the interplay between ecology and evolution in populations with differential motility. In our model, cell motility underpins group formation and allows cells to forage individually and collectively. Notably, slower cells can exploit the propulsion by faster cells within multicellular groups. When the level of such exploitation is let evolve, increasing social conflicts are associated to the evolutionary emergence of self-sustained oscillations. Akin to aggregative life cycles, resource exhaustion triggers group formation, whereas conflicts within multicellular groups restrain resource consumption, thus paving the way for the subsequent

**Data Availability Statement:** All relevant data are within the manuscript and its Supporting information files.

**Funding:** SDM has received support by the project ANR-19-CE45-0002 'ADHeC' (PSL* Research University). LM's internship was funded through

the program 'Investissements d'Avenir', launched by the French Government and implemented by ANR with the references ANR-10-LABX-54 MEMOLIFE, ANR-10-IDEX-0001-02 PSL* Research University. The funders had no role in study design, data collection and analysis, decision to publish, or preparation of the manuscript.

**Competing interests:** The authors have declared that no competing interests exist.

unicellular phase. The evolutionary transition from equilibrium coexistence to life cycles solves conflicts among heterogenous cell types by integrating them on a timescale longer than cell division, that comes to be associated to multicellular organization.

## Introduction

Multicellular life cycles have evolved multiple times during the history of life. Their emergence is thus believed to follow from general mechanistic principles, rather than from rare fortuitous events that took place in a single lineage [1, 2]. In at least six occasions, transitions to multicellularity gave rise to aggregative multicellular life cycles [1, 3], where the multicellular body forms by aggregation of dispersed cells. Such cells need not be genetically identical and can reproduce also in isolation. In aggregative life cycles, thus, conflicts withing groups [4, 5], as well as between solitary and grouped cells [6–8] appear unavoidable. Cell-level conflicts are thus predicted to hinder the evolutionary stability of collective functions—notably those achieved by division of labour between different cell types—and ultimately to doom altogether this type of multicellular organization [9]. Traditionally, theoretical models for the evolution of multicellular organization focus on such conflicts, that manifest whenever cells that invest more or less in a collective function coexist within social groups. The contribution of different types of cells is typically evaluated at the time of completion of multicellular development, after which ensues a dispersal phase. In the amoeba *D. discoideum*, for instance, strains that produce a disproportionately large amount of spores in chimeric fruiting bodies are interpreted as 'cheaters' that undermine cooperation within the multicellular structure [10]. Evolutionary game theory offers solutions to the maintenance of collective cooperative behaviour, under the assumption that groups form over and again (e.g. in the famous trait-group model [11]). For *D. discoideum*, several options have been proposed [12], that range from biasing the composition of the multicellular groups [4, 13], to modulating the individual investment in response to group composition [14, 15]. These game-theoretical explanations only focus on one specific phase of the life cycle, while they disregard the mechanisms that enable such phase to occur repeatedly. The effect of varying selective pressure that cells experience along a life cycle has instead be taken into consideration in models and experiments exploring the evolution of life cycles [16–18]. However, the time scale associated to the life cycle was extrinsically imposed and thus requires an appropriate source of environmental variation—for instance the day-night cycle—prior to the emergence of subsequent adaptations.

Here we address the emergence of a new time scale in the eco-evolutionary dynamics of populations facing a trade-off between performance in the multicellular aggregates and in isolation. Such a trade-off can occur when collective function is achieved through traits that also affect the behaviour of isolated cells. We focus in particular on differences in motility, that underpin both the benefits gained by group migration, and the capacity of single cells to feed efficiently (other possible mechanisms will be touched upon in the discussion). We show that, under selection for increased performance within heterogeneous groups, evolution leads to the emergence of an intrinsic timescale associated to the alternation of solitary and aggregated phases.

Cell motility, a widespread feature in aggregative multicellular species, is assumed here to be an ancestral trait that is heterogeneously represented in a cell population. In *D. discoideum*, amoebae exploit their individual motility to feed on bacteria in the soil. When they starve, they form multicellular slugs whose collective motility is essential to ensuring dispersal of spores that seed the following generation [19]. Cells that have lower motility before aggregation are

more likely to become exploitative spores [20]. As spores are positioned in the rear of the slug, they can benefit of the traction by more motile cells present at the front [21]. In itself, motility has the potential to drive cell-cell encounters and the consequent clustering into aggregates. At sufficient density, spatial self-organization of motile particles is known to give rise to aggregates [22]. By virtue of being part of a group, the constituent cells can reap advantages of the collective organization, for instance predation resistance and the opportunity of sharing public goods. Notably, individual motility results in enhanced directional and tactic collective displacement, which allows cells to escape the arena of local competition for space and resources [23, 24].

Although it supports collective function on the ecological timescale, cell-level motility may on the other hand destabilize cooperation within cellular collectives on evolutionary times. Firstly, motility differences within aggregates can produce conflicts for the exploitation of the benefits of collective displacement. Akin to what observed in *D. discoideum*, cells that invest more energy in displacing the group, thus providing a public good, may have a selective disadvantage. Secondly, cell motility enhances mixing in group formation, thus opposing positive assortment mechanisms, such as kin recognition, that are known to support cooperative behaviour [13]. One could therefore expect that social conflicts within heterogeneous multicellular aggregates may lead to a prevalence of slow cells, that would be unable to sustain efficient collective displacement, a scenario captured by the so-called 'tragedy of the commons' [25, 26].

We show that the picture changes when feedbacks between cell behaviour and their environment are also taken into account [27–30]. In our case, this requires considering possible advantages that motility confers to cells both in isolation and as a consequence of collective displacement. Eco-evolutionary cycles are now possible, where cells alternate phases when they are found chiefly in isolation or aggregated. Alike aggregative life cycles, grouping is triggered by depletion of environmental resources. The resulting heterogenous aggregates experience social conflict, and are eventually superseded by individually dispersing cells. These oscillations are essentially related to the population traits and have a typical timescale—longer than a cell's generation—that sets the pace of recurrence of the multicellular stage. We use adaptive dynamics to show that such emergent timescale arises in the course of evolution as selection increases the intensity of social conflicts. In the discussion, we address the possibility that cycles of aggregation, coupled to the demography of cells of different motility, may act as a scaffold to the evolution of aggregative life cycles.

## Models and methods

### Eco-evolutionary model for fast and slow cell types consuming a shared resource

We describe the 'ecological' (resource-consumer) dynamics of a population of $N$ cells, coupled to the 'evolutionary' variation of the frequencies of two types of cells, that differ in a heritable motility trait: a fraction $x$ of cells is fast-moving and a fraction $(1 - x)$ is slow-moving. Both cell types forage on a shared resource of density $R$. The resource is assumed to grow logistically in the absence of consumption, and is consumed at the same rate by all cells. Instantaneous growth rates of each cell type depend on the product of resource density and reproductive efficiency, as measured by 'payoffs' $p_F$ and $p_S$—discussed below—that take into account the partition between the solitary and aggregated phases of the fast and slow type respectively. The cell population size thus changes at a rate equal to the the average payoff $\bar{p} = x \, p_F + (1 - x) \, p_S$ times the resource density. Corresponding to these payoffs, the frequencies of cells with the two motility traits change in time according to a replicator equation for the fraction $x$ of fast cells [30, 31]. The time variation of the resource density, total cell population size and fraction

of fast cells is thus described by the following set of ordinary differential equations:

$$\frac{dR}{dt} = R\left[r\left(1 - \frac{R}{K}\right) - N\right]$$

$$\frac{dN}{dt} = N\left[\bar{p}(x, R)\,R - d\right] \qquad (1)$$

$$\frac{dx}{dt} = x\,(1 - x)\,R\left[p_F(x, R) - p_S(x, R)\right]$$

where $r$ and $K$ are the maximum growth rate and the carrying capacity of the resource, respectively (rescaled so that the probability of encounter between the resource and the cells is 1 in a time interval), and $d$ is the mortality rate of cells (assumed to be identical for every cell type). These equations can be equivalently formulated in terms of resource density and the number of cells of slow and fast types, as shown in S1 Text. Contrary to the sole replicator equation, they describe the dynamics of both cell population composition and size, beside that of resource density. We consider that payoffs of the two cell types (illustrated in Fig 1) depend on environmental conditions [28] through the resource density $R$. Moreover, they depend on the social context, reflecting differences in foraging efficiency and social investment when cells are either in isolation or inside groups.

The payoffs represent the success of different motility types in competition inside or outside groups. First of all, we consider that the partition of the population between isolated cells and groups depends on resource availability. We suppose that when cells are occupied in food acquisition and handling, they tend to keep feeding locally. We model this by assuming that, at any point in time, the fraction of cells that is in a group (whether fast or slow) is negatively correlated with resource density, so that all cells are solitary when the resource is at carrying capacity, and that all cells are in groups when the resource is completely depleted. We also assume for simplicity that groups are formed by randomly drawing cells from the population.

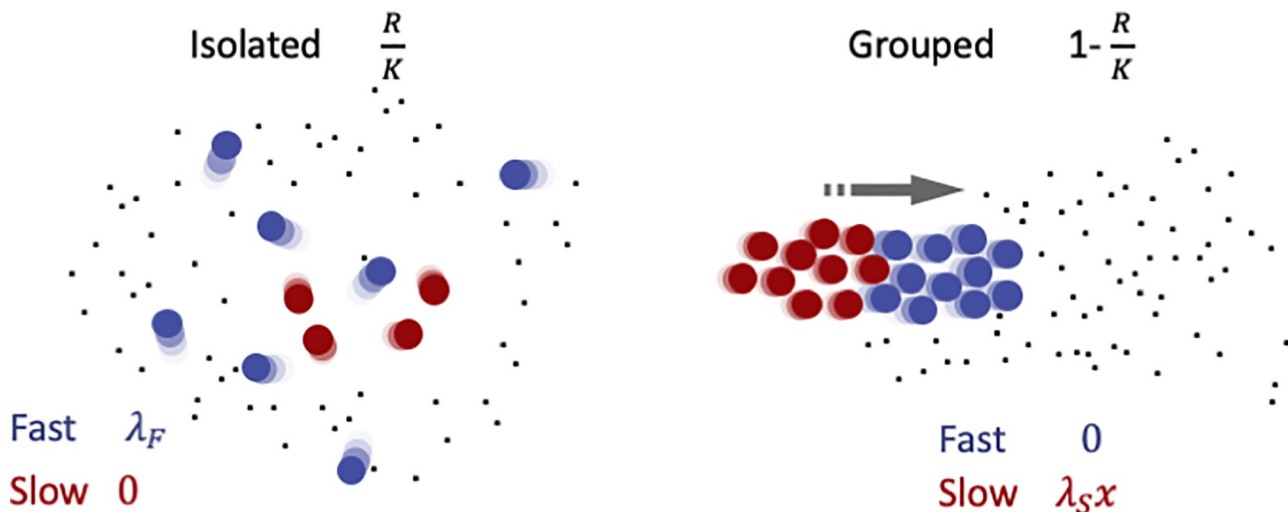

**Fig 1. Illustration of payoffs of slow (red) and fast (blue) cells in different social contexts.** The probability of cells to be found alone (left) or grouped (right) depends on the resource density and is indicated on top. On the left, cells displace individually in a patch of resource. Slow cells are disadvantaged with respect to fast cells in local foraging, as indicated by the context-dependent payoffs below (explained in detail in the main text). The right panel illustrates the alternative scenario, when cells are grouped. Collective displacement toward new patches of resource is fuelled by fast cells. Slow cells take now advantage of the new resource patch and thus exploit fast cells, that instead reap no benefit, having spent all their energy in propulsion. Within aggregates, thus, slow cells behave as social cheaters in a public goods game.

The average ratio of cells between types within a group will thus be, independently of the group size, equal to the proportion of the types in the population. Note that this proportion, as well as the actual number of fast and slow cells inside and outside aggregates, can vary in time. Then, we reason that while isolated cells compete locally for nutrients, cells inside groups can take advantage of collective behaviour for escaping local overcrowding more efficiently than individual cells. Formation of heterogenous aggregates at the same time creates disruptive conflicts as to the contribution to collective displacement. Computation of the average payoffs of cells of any type thus requires separately evaluating performances in isolation and within groups. Fast cells are efficient in looking for immediately available food, and outcompete in this task slow cells. Whenever moving a short distance is sufficient to reach food items, fast cells indeed thrive in isolation. We therefore assume that the probability that cells remain outside aggregates is proportional to the amount of food available in the environment. We consider for simplicity that this probability is $\frac{R}{K}$, that is cells are always alone when the resource reaches its carrying capacity (when the proportionality factor is different, the results are qualitatively the same, as discussed in S2 Text). The payoff of isolated fast cells will thus be $\frac{R}{K}\lambda_F$, where $\lambda_F$ measures how efficient they are in solitary feeding. Within groups, instead, fast cells spend all their energy in propelling the aggregate, including slow cells, so that they have a null payoff. Slow cells, conversely, cannot reproduce in isolation, because they are inefficient at chasing local resource items. They can however benefit of the collective displacement of the group, whose propulsion is sustained by fast cells. As in classical public goods games that model social conflict, benefits of the multicellular organisation reaped by slow cells are assumed to increase with the fraction of fast cells in the groups [32]. This is the product of the fraction of fast cells in the population times the probability that a fast cell is in a group $\left(1 - \frac{R}{K}\right)$. The average payoff of slow cells is then $x\left(1 - \frac{R}{K}\right)\lambda_S$, where the parameter $\lambda_S$ measures the ability of slow cells to exploit fast cells within the collective phase in order to enhance their own success. $\lambda_S$ will be later considered as an evolvable trait. The average payoff in the population is then:

$$\bar{p}(x, R) := p_F\, x + p_S\,(1 - x) = \left[(\lambda_F - \lambda_S)\,\frac{R}{K} + \lambda_S\right] x - \left(1 - \frac{R}{K}\right)\lambda_S\, x^2. \qquad (2)$$

A summary of the parameters involved in the model is shown in Table 1. Our choice of the payoffs is an extreme case of more realistic scenarios when fast cells can also reproduce within groups, and slow cells in isolation. We have chosen this simple form because it exemplifies the hardest possible social conflicts, those associated with the death of one of the cell types, as observed in some extant species of Dictyostelids (e.g. *D. discoideum*). It also allows for more straightforward analytical solution. However, numerical simulations (not shown) indicate that qualitatively similar results hold when both cell types reproduce both inside and outside groups, as long as a sufficiently intense trade-off exists between the benefits of movement in isolation and those gained by social displacement.

**Table 1. List of the parameters used in the model.**

| Parameter | Description |
|:---:|:---:|
| $d$ | Consumer death rate |
| $K$ | Resource carrying capacity |
| $r$ | Resource maximum growth rate |
| $\lambda_F$ | Fast cell payoff |
| $\lambda_S$ | Slow cell payoff |

## Results

### Different motility types can coexist in equilibrium or organize along aggregative cycles

We first address how cell population partitioning in grouped and solitary components and their composition in fast and slow types change in time, for a fixed set of parameters. Among the possible eco-evolutionary regimes available to the cell population, we are particularly interested in self-sustained oscillations and in their associated timescale. Later, we will explore how such dynamics change when the parameter defining the intensity of social conflicts can evolve. The model described by Eq (1) has two qualitatively different dynamical regimes of coexistence between fast and slow cells: a stable equilibrium and a stable limit cycle (a detailed analysis of all the equilibria and their stability is provided in S1 Text). When the coexistence equilibrium $(\hat{R}, \hat{N}, \hat{x})$ (present whenever $\sqrt{\lambda_F K / d} > 1 + \lambda_F / \lambda_S$) is stable, slow and fast types are found in constant proportions both as free and aggregated cells. The two cell types survive because of their respective advantages in one or the other state of aggregation. Neither the total population size, nor resource availability change in time, so that there is no timescale associated to demography or cell type frequencies. When the coexistence equilibrium becomes unstable, it is surrounded by a stable limit cycle (Fig 2), that we call 'life-like cycle'. In the oscillating regime, indeed, the eco-evolutionary dynamics has a temporal structure akin to the life cycle of aggregative multicellular organisms, with an environmentally-triggered alternation of solitary and grouped stages (Fig 2). The population size undergoes limit-cycle oscillations, where the total number of cells has a phase delay with respect to the resource (as in classical resource-consumer ecological models). At the meantime, the proportion of cells that are found within groups and the composition of both grouped and solitary fractions change in time (S1 Fig): when resources are abundant, a small percentage of cells is grouped and fast cells grow in number by virtue of the advantages gained by feeding locally; as resources are progressively exploited, more and more cells are found inside groups. Slow cells can thus exploit the contribution of fast ones towards collective displacement, so that 'cheating' increases in the population (S2 Fig). By overthrowing the collective function, the 'tragedy of the commons' causes the overall payoffs, thus the population growth rate, to decline. Lower consumption now allows resources to build up again, providing renewed opportunities for fast cells to multiply in isolation.

Although consumer oscillations have a similar phase arrangement as predator-prey ecological dynamics, they hinge upon the coupling of ecology and evolution. Purely ecological equations can be obtained in the neutral case when both types have the same payoff, so that the frequency of fast and slow cells is constant. The demographic dynamics is then described by the first two equations, with the average payoff being evaluated for that fixed frequency (S3 Text). In this case, linear stability analysis shows that the coexistence equilibrium is always asymptotically stable. In order to determine in what circumstances equilibrium coexistence or non-steady behaviour should be expected, we examine the dependence of the dynamic regimes of Eq (1) on parameters, focusing in particular on those defining the payoffs of fast and slow cells. In the plane $(\lambda_S, \lambda_F)$, limit cycle oscillations (Fig 3A, inset) appear when the coexistence equilibrium loses its stability as a result of a supercritical Hopf bifurcation. We can use the bifurcation condition (implicitly defined as the solution of a third degree polynomial, see S1 Text) to delimit the region in parameter space where the dynamics is out-of-equilibrium. Fig 3 displays the amplitude (panel A) and period (panel B) of the life-like cycle, respectively. The parameter-dependence of the dynamics is best illustrated when one parameter at a time is let vary. The bifurcation diagram when the level of exploitation $\lambda_S$

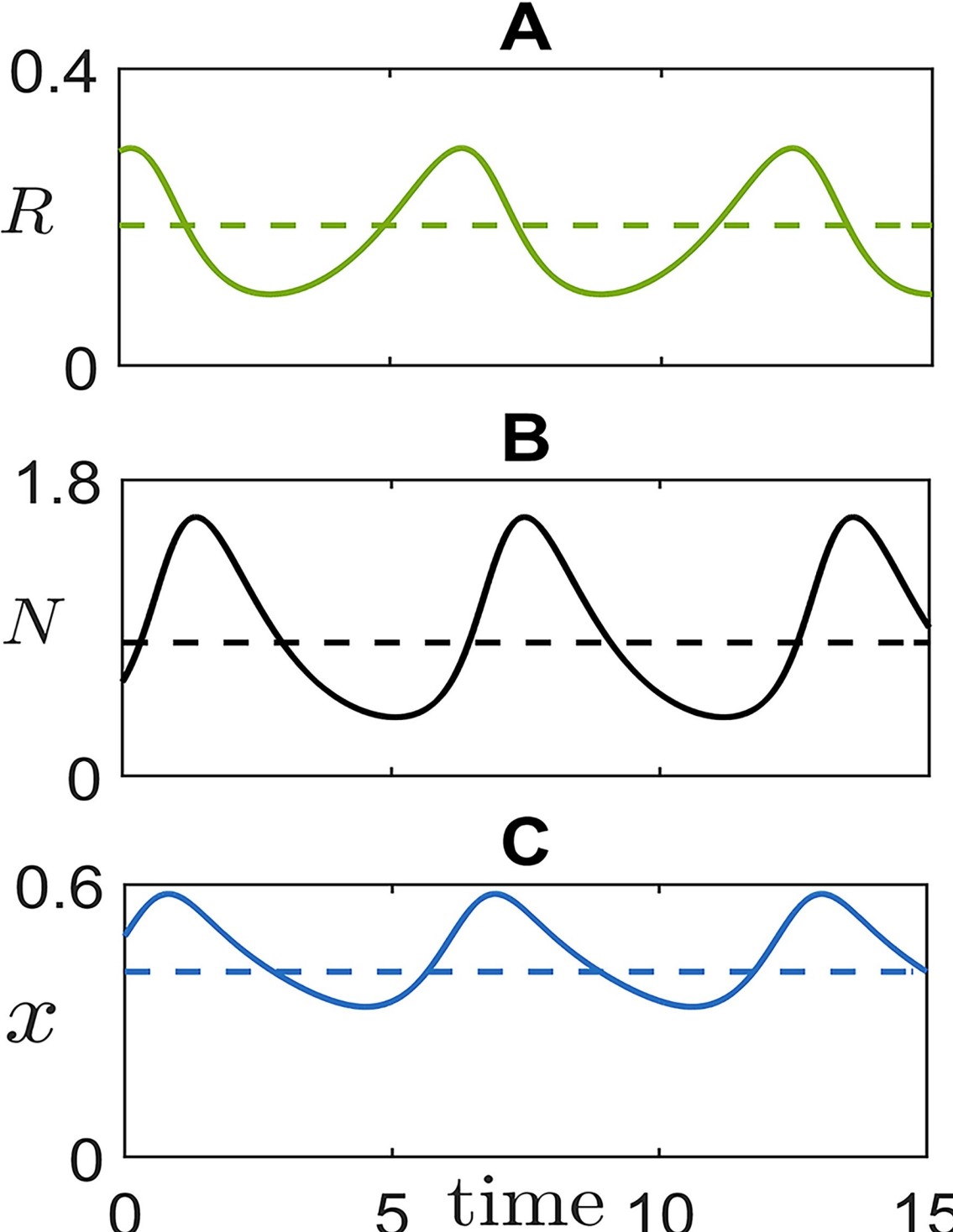

**Fig 2. Eco-evolutionary cycle of cells of differential motility and their resource.** Representative oscillations of resource concentration, total population size and fraction of fast cells. Dashed lines indicate the position of the unstable coexistence equilibrium (analytically derived in S1 Text). The simulated trajectory illustrates the temporal arrangement of cycles in the model, similar to that of aggregative life cycles: aggregation is triggered by resource depletion; aggregates provide the collective function, but offer to slow cells the opportunity for social exploitation; as group function is degraded by the rising of 'cheaters', resources can build up anew, restarting the cycle. Parameter values are: $r = 1$, $K = 1$, $d = 1$, $\lambda_F = 28$, $\lambda_S = 16$.

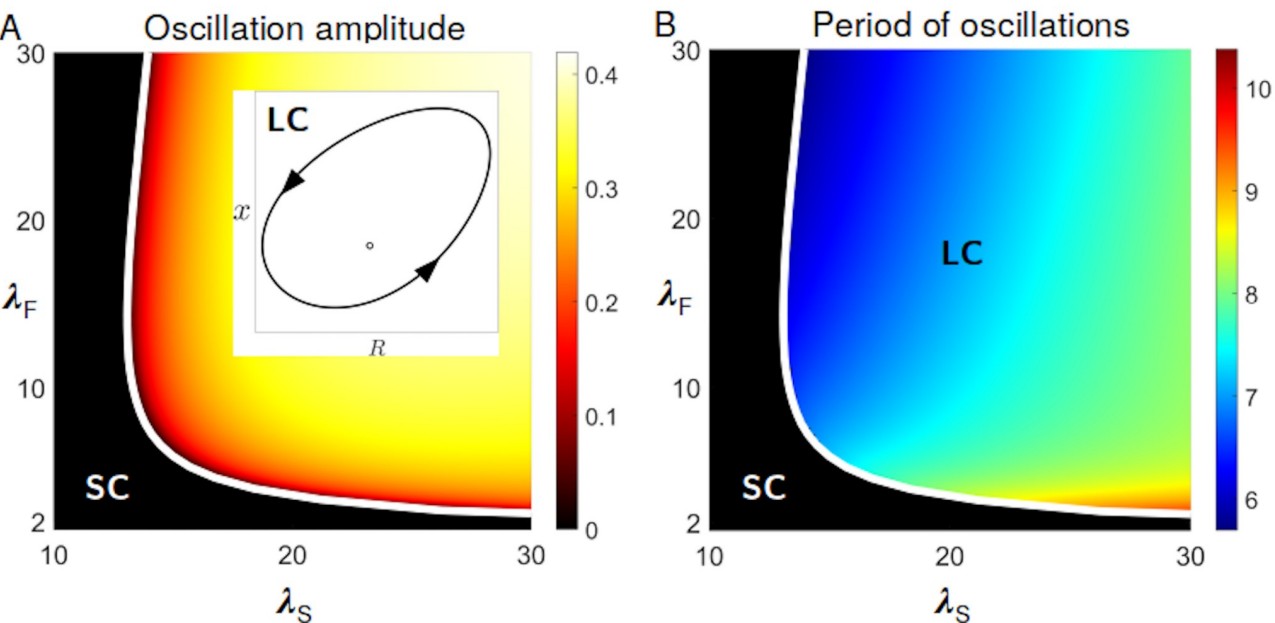

**Fig 3. Dynamical regimes and dependence on cell-level parameters.** Eq (1) display two main qualitatively different dynamics: limit cycle oscillations (LC), where an unstable equilibrium coexists with a stable limit cycle, as illustrated in the inset of panel A, and a stable coexistence equilibrium (SC). The bifurcation diagrams in panels A and B recapitulate the dependence of the eco-evolutionary dynamics on the two parameters determining the benefits at the cell level: strength of social exploitation by slow cells $\lambda_S$, and advantage of solitary living for fast cells $\lambda_F$ (other parameters are as in Fig 2). The heatmaps reveal the variation of amplitude (A) and period (B) of the oscillations. The white line (analytically derived in S1 Text) indicates the Hopf bifurcation where the coexistence equilibrium changes stability. In the oscillating region, the timescale associated to the life-like cycle is slower than those associated to cell division (see S5 Fig).

is the control parameter, and the others are held constant, is illustrated in S3 Fig. When exploitation is low, fast and slow cells coexist. Past the bifurcation point, demographic oscillations become increasingly large and slow, but they do not appear to approach, as exploitation becomes more severe, a global bifurcation that would break down the oscillations (S4 Fig). Correspondingly, as social conflict increases, grouping becomes more and more associated to a specific phase of the life-like cycle. Oscillations in the fraction of fast cells remain instead bounded because their success comes with their doom: the more numerous they are, the more they get exploited by slow cells (S3 Fig).

A characteristic feature of aggregative life cycles, and more broadly of multicellular organization, is that the duration of the higher-level cycle encompasses several cellular generations. Therefore, we compared the period of the aggregation life-like cycle with two timescales associated with cellular demography: the maximal and average growth rate of the two cell types during population-level oscillations. In both cases (S5 Fig) the period of the cycle is longer than the duplication time of the cells, and can thus be consistently interpreted as the duration of collective-level generations. The time-scale separation is highest close to the onset of the oscillations (see S5 Fig). As the amplitude of the oscillations increases, the period of the life-like cycle decreases, but it always remains larger than a cell generation. We have seen that the feedback between ecology and population composition can give rise to a temporal compartmentalization of cell behaviour along a cycle, with slow and fast cells taking advantage alternatively of collective and individual motility. Next, we examine whether such a life-like cycle can be expected to emerge and be maintained when the key cell-level parameter responsible for social cheating—the level of exploitation of the collective function by slow cells—is allowed to evolve.

## Evolutionary increase of exploitation intensity drives the emergence of life-like cycles

In multicellular organisms, cheating by lower-level, independently reproducing cells is expected to destabilize the collective function [25, 26]. In the framework of our model, cheating occurs when slow cells exploit fast cells for propulsion. If competition between fast and slow cells occurred exclusively within social groups, then the slow type would invade, and eventually cause the decline of collective movement. We consider now that the social exploitation parameter $\lambda_S$ is a continuous trait subjected to mutation and selection, and study its evolutionary changes in the framework of adaptive dynamics [33]. Long-term variations in the exploitation level occur as a resident population is repeatedly challenged by mutants with a different trait value. If an infinitesimally small number of mutants grows in frequency (i.e. has positive invasion fitness), the new trait is assumed to substitute the resident. Numerical simulations in the oscillation region confirmed that such substitution actually occurs, and coexistence of the mutant and the resident was never observed (not shown). Invasion fitness of a mutant needs to be evaluated by considering that the total population is composed of three components: fast cells, resident slow cells and mutant slow cells that differ only in their value of $\lambda_S$. The dynamics is thus described by five ODEs with the constraint that frequencies add to one (S4 Text). When the cell types coexist at equilibrium, computation of the growth rate of a rare mutant of trait $\lambda_S$ into a resident population with parameter $\lambda_S^*$ yields:

$$S(\lambda_S, \lambda_S^*) = d \, \frac{\lambda_S - \lambda_S^*}{\lambda_S^*}. \tag{3}$$

The invasion fitness $S$ thus has the same sign as the difference in exploitation level between the mutant and the resident. Once the system has transitioned to a cyclic regime, it is not possible to compute the invasion fitness analytically. In order to find if the evolutionary dynamics pushes $\lambda_S$ consistently towards higher values, or a reversal in the direction of evolution happens in the non-equilibrium regimes, we estimated numerically the invasion fitness as the average rate of increase in the frequency of an initially rare mutant type. Fig 4 displays the invasion fitness for different values of the resident exploitation parameter $\lambda_S^*$, assuming that mutations produce a small increment in exploitation ($\lambda_S - \lambda_S^* = 10^{-1}$). In both equilibrium and cyclic regimes, invasion fitness remains positive for all levels of exploitation. This means that exploitation becomes progressively more severe, as one might expect given the advantages of cheaters within social groups. Nonetheless, such evolutionary change also drives the system toward the bifurcation point, so that as cheating becomes more effective, the emergence of the new collective timescale generates a temporal compartmentalization, whereby social conflicts dominate in only one phase of the cycle. Moreover, invasion fitness scales as the reciprocal of $\lambda_S$ (Fig 4, inset), therefore the time necessary for a mutant with increased $\lambda_S$ to invade the population grows progressively larger. The population might therefore reach the limit when the timescale of mutations is comparable to that of trait substitution, opening the door to a possible quasi-neutral coexistence, along an oscillatory trajectory, of strains with different levels of exploitation. This could offer an explanation, alternative to limited dispersal and fast evolutionary variation [29], to the observation that coexistence of different strains rather than competitive exclusion seems to characterize natural cell populations. In aggregative microbes, an unrestrained escalation of exploitation levels may hence boost genetic diversity by diminishing the returns to cheating. At the same time, however, the system would be driven toward regimes with higher excursions in population size, where stochastic fluctuations in finite populations may cause the population to go extinct.

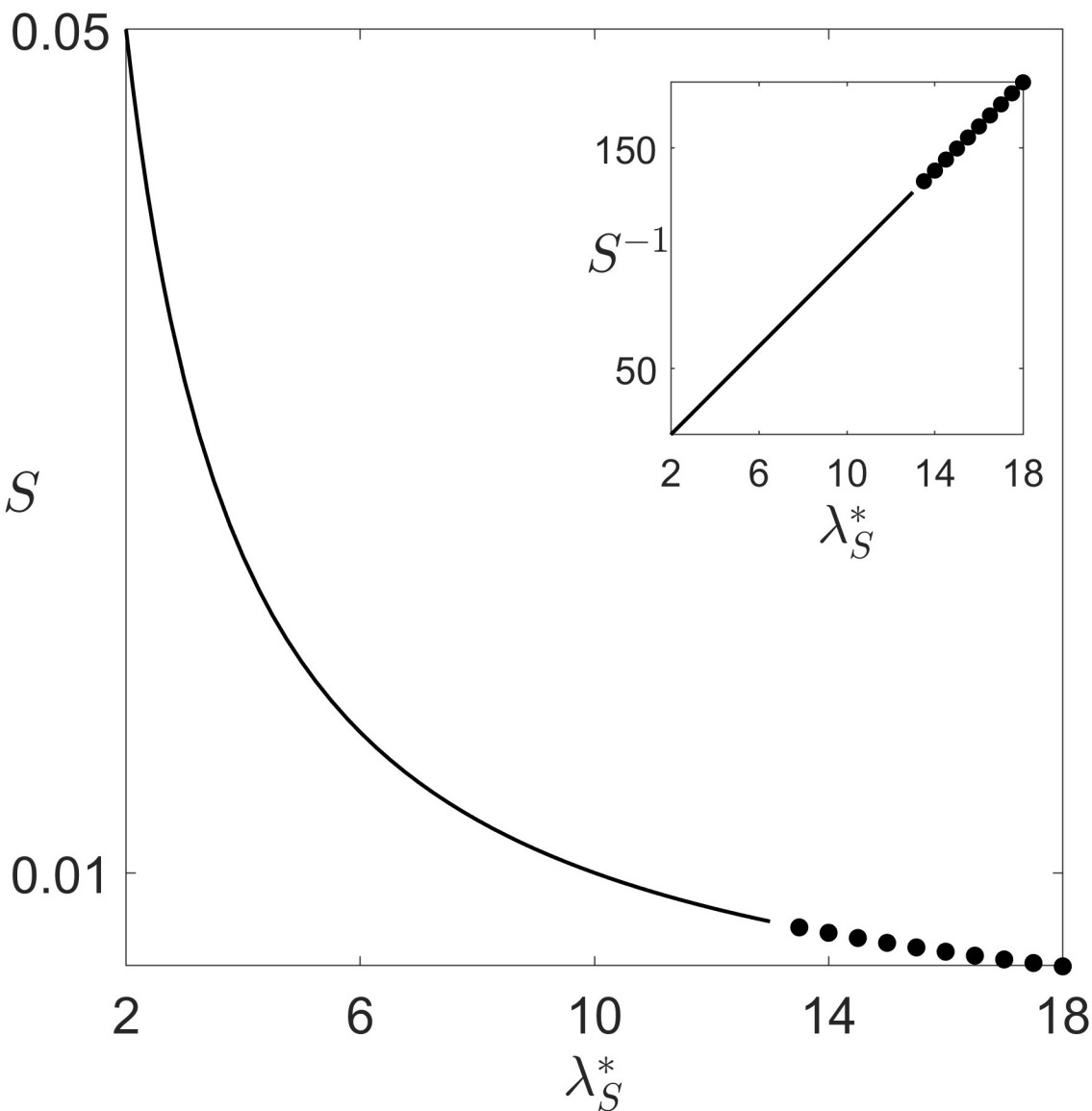

**Fig 4. Invasion fitness.** Invasion fitness of mutants whose level of social exploitation $\lambda_S$ is higher with respect to the resident trait $\lambda_S^*$ as a function of the latter (remaining parameters are as in Fig 2). Continuous lines represent Eq (3), dots the rate of increase of perturbations transverse to the limit cycle, averaged over a period of the cycle (S4 Text). The invasion fitness is always positive, leading to selection of ever-increasing levels of cheating, but its decline means that the evolutionary dynamics of the trait grows progressively slower.

## Discussion

The emergence of life cycles involving a multicellular stage is a necessary step in major transitions in individuality, whereby organization at higher levels provided access to novel collective functions [2]. The evolutionary origin of collective reproduction and life cycles has been addressed both theoretically [27, 34–36] and experimentally [17, 36] in systems where single-cell bottlenecks and clonal expansion ensure efficient purging of cheating types. The evolutionary establishment of aggregative life cycles, where different cellular types can come together, has instead been much less explored. Here, we have considered the possible role of pre-existing

differences in cell motility in the evolution of eco-evolutionary 'life-like' cycles, whereby cells come to alternate aggregated and solitary phases on a nascent timescale, on which both population size and composition undergo periodic oscillations. Such a timescale is longer than that of cell division and emerges as an adaptive response to social conflicts that differential motility raises within multicellular aggregates. Setting the pace for recurrence of heterogenous aggregates, it can be associated to a higher level of cellular organization, and act as a scaffold to subsequent evolutionary innovation [18]. Alternating selective pressures are considered a basic ingredient for the emergence of multicellular life cycles. They can be extrinsically forced by environmental fluctuations, either periodic or stochastic, that exogenously set the timescale over which benefits and costs of multicellular organization are evaluated [37, 38]. In our model, equilibrium coexistence of cells with different motility evolves into a cycle that alternates phases where either fast or slow cells are selectively advantaged. The timing of these phases depends on cell-level parameters, and changes along the evolutionary trajectory. As in the case of extrinsic oscillations, such alternating selective pressures, generated by genetically distinct partners, may set the scene for selection of more complex strategies for cell behaviour, notably phenotypic switching [39] or context-dependent phenotype determination [15, 16, 40]. The possibility that extrinsically imposed periodic changes in selection lead to the emergence of phenotypic variation typically associated to life cycles has been experimentally demonstrated in Hammerschmidt et al. [17]. In our work, we thus focused on the emergence of the timescale of such variation. Extensions of the model to cases when behaviour is not or is only partially heritable would however be a natural next step towards the complete integration of cell ecology in the evolution of aggregative life cycles. We have shown that increasing social conflicts drive the transition from an initial state, where different conflicting types coexist at equilibrium, to eco-evolutionary limit cycles, where their frequency is coupled to the partition of cells between solitary and aggregated states. Such cycles bear numerous analogies to aggregative life cycles: they display recurring phases of enhanced aggregation, and are characterized by conflicts within aggregates, as well as between aggregates and solitary cells. When resources are scarce, fast cells behave as cooperators fuelling collective motion [4, 21]. However, they play the role of 'loners' by feeding on locally available resources when these are plentiful [7, 8, 41]. Cooperation therefore stands as a side effect of individual strife for survival, which only manifests when fast cells join groups by chance [42]. Slow cells, on the other hand, cheat within groups but cannot survive in the absence of fast cells, so that social exploitation curtails itself through population dynamics. When cheating increases—as predicted by the 'tragedy of the commons' [25, 26]—over evolutionary times, the system progressively moves towards the oscillatory regime, where conflicts are solved through the temporal compartmentalisation of social investment. Previously, non-steady behaviour was identified as a means to maintain cooperation in spite of cheaters success within groups [28, 30, 43–45]. Individual-based simulations taking motility into account showed that limit cycle oscillations in the frequency of cooperators and cheaters could occur due to the undermining of the collective function by the spread of cheaters. Their period encompassed numerous collective cycles, each marked by a discontinuous dispersal event (whereby cells were randomly reallocated in space) [32, 46]. In the model discussed here, the timescale over which social conflicts play out is instead the same as that of the aggregation dynamics, and defines—independently of the initial state of the population—the period of recurrence of the collective state. It is therefore more appropriate to describe the origin of the temporal organization of life cycles, rather than a specific cycle already punctuated by a single dispersal event (which could instead represent a successive adaptation). An indication that motility phenotypes may have been involved in the emergence of aggregative life cycles is that the emergent eco-evolutionary cycles bear many similarities to the life cycle of *D. discoideum*. In particular, our model depicts a continuous exploitation of

fast cells by slow cells, that occurs not only at the stage of reproduction (as in game-theoretical models), but at any step of the multicellular phase, analogous to what had been proposed to occur in slime moulds [21]. Even though the example of *Dictyostelium* made us focus on motility as the factor underpinning collective function and conflicts among different cell types, we expect similar transition to take place when conflicts and trade-offs stem from other heritable traits that affect both solitary living and collective function, notably from differences in adhesion or in sensitivity to signals. In order to distinguish among different possible cell-level features in their effect on the population cycles, more realism needs to be introduced in the description of the population dynamics. Simple deterministic equations allowed us to fully characterize the parameter-dependence of the dynamic regimes, and to apply analytically adaptive dynamics theory. Individual-based models however would offer the opportunity to examine more closely other aspects of the eco-evolutionary dynamics, such as demographic fluctuations and group formation [24, 47, 48]. Finite-size fluctuations [49–51], in particular, are expected to be important if cell-level parameters attain, along an evolutionary trajectory, regions where the population bottleneck becomes more extreme. By assuming that fast and slow cells have the same probability of being found inside groups, our model describes in a very crude way the process of group formation. Differences in speed, possibly associated to differences in adhesion, may indeed induce differential grouping properties among cell types. Since traits that influence assortment affect the evolutionary process [13, 23, 46, 48, 52–54], a more detailed description of motility-induced biases would be important to evaluate the applicability of our conclusions to specific microbial populations. In particular, future studies describing explicitly the process of group formation may address the consequences of evolution of motility on dispersal strategies [55–57].

## Supporting information

**S1 Text. Equilibria of the eco-evo dynamics and linear stability analysis.**
(PDF)

**S2 Text. Bounded probability of remaining alone.**
(PDF)

**S3 Text. Linear stability analysis of the purely ecological dynamics.**
(PDF)

**S4 Text. Adaptive dynamics.**
(PDF)

**S1 Fig. Eco-evolutionary limit cycle oscillations in the number of aggregated and isolated cells.** The numbers of fast and slow cells found in the solitary and in the aggregated state oscillate in time along a life-like cycle. These quantities are computed along a limit cycle solution of the eco-evolutionary dynamical system, for the same parameter values as Fig 2 of the main text.
(TIFF)

**S2 Fig. Payoffs of the fast and slow cell types in the oscillating regime.** The payoffs of the two strategies oscillate in time as a consequence of the variations in the fraction of aggregated cells and in group composition. These quantities are computed along a limit cycle solution of the eco-evolutionary dynamical system, for the same parameter values as Fig 2 of the main text.
(TIFF)

**S3 Fig. Bifurcation diagram for varying strength of social exploitation by slow cells.** Bifurcation diagrams of the three state variables as a function of the exploitation parameter $\lambda_S$ (remaining parameters as in Fig 2 of the main text). Continuous lines indicate the stable coexistence equilibrium and the stable limit cycle, the dashed line indicates the unstable equilibrium. The transition from the coexistence equilibrium point to the limit cycle occurs through a supercritical Hopf bifurcation. The parameter values that identify this bifurcation are numerically computed as explained in S1 Text and are illustrated by the white line in Fig 3 of the main text and S5 Fig.
(TIF)

**S4 Fig. Period of oscillations against exploitation strength.** The period of the eco-evolutionary limit cycle increases a function of the exploitation level $\lambda_S$ (remaining parameters as in Fig 2 of the main text). The fact that the period does not diverge indicates that an increase in exploitation level does not drive the system towards a global bifurcation that would go undetected by the local analysis of the equilibria we performed.
(TIF)

**S5 Fig. Ratio between the timescale of the life cycle and (maximal/average) cell generation length.** Bifurcation diagrams with respect to the parameters $\lambda_S$ and $\lambda_F$ (remaining parameters as in Fig 2 of the main text) superimposed to: A) the numerically obtained heatmap of the ratio between the period of the limit cycle and the *fastest* timescale of the demographic dynamics, computed as the maximum value of the population growth rate along the limit cycle; B) the numerically obtained heatmap of the ratio between the period of the limit cycle and the *mean* timescale of the demographic dynamics, computed as the average value of the population growth rate along the limit cycle. The white lines are the bifurcation curve analytically derived in S1 Text. The ratio being consistently larger than one indicates that the eco-evolutionary dynamics (thus the cycle of aggregation) occurs on a time scale that is slower than that of cell-level reproduction.
(TIF)

## Acknowledgments

LM thanks RML Evans and Robert West for fruitful discussions. SDM is grateful to Paul Rainey, Sandrine Adiba and Mathieu Forget for inspiring discussions, and to Alice l'Huillier and Bertrand Maury for the use of Cardano's formula in the semi-analytical derivation of the bifurcation curve. LM and SDM are thankful to Yuriy Pichugin, Allyson Sgro for their insightful comments on the manuscript.

## Author Contributions

**Conceptualization:** Leonardo Miele, Silvia De Monte.

**Formal analysis:** Leonardo Miele, Silvia De Monte.

**Funding acquisition:** Silvia De Monte.

**Investigation:** Leonardo Miele, Silvia De Monte.

**Methodology:** Silvia De Monte.

**Project administration:** Silvia De Monte.

**Resources:** Silvia De Monte.

**Software:** Leonardo Miele.

**Supervision:** Silvia De Monte.

**Validation:** Leonardo Miele, Silvia De Monte.

**Visualization:** Leonardo Miele.

**Writing – original draft:** Leonardo Miele, Silvia De Monte.

**Writing – review & editing:** Leonardo Miele, Silvia De Monte.

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
