## [Decision Letter · Decision Letter 0]

2 Sep 2020

Dear Mr. Miele,

Thank you very much for submitting your manuscript "The evolutionary emergence of aggregative multicellular life cycles" for consideration at PLOS Computational Biology. As with all papers reviewed by the journal, your manuscript was reviewed by members of the editorial board and by several independent reviewers. The reviewers appreciated the attention to an important topic and were unanimously enthusiastic about the work. Based on the reviews, we are likely to accept this manuscript for publication. However, note that two of the reviewers raise fundamental questions about the biological interpretation of the (otherwise very interesting) population-level cycles as life cycles. This aspect needs to be thoughtfully and thoroughly addressed in the manuscript before we can accept it for publication.

Sincerely,

Corina E. Tarnita

Associate Editor

PLOS Computational Biology

Stefano Allesina

Deputy Editor

PLOS Computational Biology

[LINK]

Reviewer's Responses to Questions

**Comments to the Authors:**

Reviewer #1: N/A

Reviewer #2: This paper concerns a topic that has recently seen increased interest: the evolution of multicellular life cycles. The authors build a model of a cell population that consumes a resource and forms aggregations when resources are low. Cells exist with different motilities: there is a “fast” type that is successful when solitary, and a “slow” type that is successful when it is in group, through the exploitation of “fast” group members. The authors show that fast and slow types can coexist, and, moreover, that the model can give rise to oscillatory behavior in which the cycles between abundant resources and mostly fast, solitary cells on the one hand, and scarce resources and mostly slow, grouped cells on the other. These oscillations are interpreted population as a mechanism for the eco-evolutionary emergence of aggregative multicellular life cycles.

This is an elegant and creative study, in which the authors use a simple, analytically tractable model to study how social exploitation affects population dynamics. The theoretical result that in a population of microbes that engage in collective behavior (i.e., collective motion), the evolution of social exploitation can lead to population-wide fluctuations is novel and interesting. The paper takes a deliberate approach in which there is no way for exploited cells to circumvent social exploitation. Exploitation of fast cells by slow cells is inevitable, which allows the authors to study the ecological consequences of such exploitation: as exploitation gets more and more severe, population-wide fluctuations emerge, meaning that resource depletion and group formation, followed by resource recovery and return to the solitary stage, become synchronized at the population level. Most of the results are obtained analytically.

My main concerns are with the presentation and the interpretation of the results. In terms of presentation, the figures can be improved (specific suggestions below). In terms of interpretation, I am concerned with the interpretation of population-wide fluctuations as life cycles. I don’t think the term life cycle applies here, and I don’t think the authors’ model captures the evolutionary emergence of a life cycle.

I have collected some specific comments below. These develop the concerns mentioned above and contain numerous suggestions that I hope will help the authors in improving the paper.

1. The authors use the term “life cycle” to describe population-wide fluctuations of fast and slow cells, as well as the resource. There is no genealogical relationship between fast and slow cells. In the absence of such a genealogical relationship, it seems inappropriate to apply the term “life cycle” here. I don’t know of examples in the literature where the term “life cycle” is used this liberally, and I am worried that this departure from the common meaning of the term would lead to confusion.

2. The model builds in the assumption that, depending on resource abundance, cells are solitary or belong to groups. I think of this setup as a dynamic equilibrium between cells and groups, in which groups form by aggregation (when resource levels drop), but groups also can give rise to single cells (when resource levels increase). These continuing dynamics are also necessary to ensure that group composition reflects the population composition of fast and slow cells, even while in groups it is only the slow cells that reproduce. The existence of these group dynamics implies that there already exists a life cycle at the very beginning of the model, which depends on the rates at which cells join and leave aggregates. In line 25, the authors write that they address “the evolutionary origin of ... life cycles from cellular populations that initially had an essentially unicellular lifestyle”—saying “essentially unicellular” seems inappropriate. I would say that the authors’ model captures that social exploitation can lead to population-wide fluctuations in the context of a pre-existing primitive multicellular life cycle, not the de novo emergence of such a multicellular life cycle. In fact, I don’t think a model that accounts for population dynamics but not for the dynamics of group formation and group propagation can say too much about the emergence of a multicellular life cycle. All this is not meant to invalidate the authors’ results; I think the model as constructed is elegant, cleverly designed in the way it captures important aspects of the biology while still being simple enough to be analytically tractable, and I think the main result is interesting. But I don’t think it should be framed as the emergence of a life cycle.

3. On a related note, I’m curious about the following. Say I build the same model as the authors, but with just a single cell type (no social exploitation), but I make slightly different assumptions so that a limit cycle emerges (this is relatively straightforward in a producer-consumer model, for example, by changing the functional response). Would the authors also interpret this as the emergence of a multicellular life cycle, because it leads to population fluctuations in the fraction of cells that belong to groups?

4. In Figure 2, please have the y axes start at 0, and, for the variable x, have it end at 1. Extend the horizontal axis so that more than a single oscillation is shown. Please also plot the fraction of cells in aggregates (SF1a), as well as the payoffs of fast and slow cells. I am confused by SF1B,C. For example, according to the caption, SF1C shows the conditional probability that a solitary cell is fast—but, according to my understanding of the model, this should just be equal to x, as being solitary is independent from being fast. The authors could also consider plotting the number of solitary slow cells, solitary fast cells, grouped slow cells, and grouped fast cells over time in a single stacked area plot to make the population dynamics for insightful for the reader.

5. In Figure 3, should the y-axis for 3B say “\\lambda_F” instead of “F”? In the inset in 3A, please indicate the direction (clockwise or counterclockwise).

6. Figure 4: should it be \\lambda_S* on the x axis? Otherwise, I don’t understand how Fig. 4 relates to equation (3), which shows that S (not S^-1) is a linear function of \\lambda_S. In Figure 4, what is the assumption on the mutant growth rate? S depends on both \\lambda_S and \\lambda_S*, but it is plotted as a function of only one.

7. How sensitive are the results to the assumption that the payoff of slow cells scales linearly with the fraction of fast cells?

8. I was initially confused by the model setup (1), in which the replicator equation is used to write an equation for dx/dt. It was not immediately intuitive to me how the equations for dN/dt and dx/dt would interact. More intuitive to me would be to, instead, write equations for the fast cells F and the slow cells S. These equations are simply dF/dt = RFp_F – dF and dS/dt = RSp_S – dS. I think it would be valuable to mention this equivalent setup as well, as it allows the reader to understand the dynamics of a single cell type from a single equation (rather than having to combine the equations of dN/dt and dx/dt).

9. In the Introduction, the grouping mechanism (lines 76-80) should be explained better. In particular, the sentence “We model this by assuming that the probability to join a group is inversely proportional to resource density” is confusing. Perhaps better is something along the lines of “We assume that, at any point in time, the fraction of cells that is in a group (whether fast or slow) is equal to 1-R/K. This means that all cells are solitary when the resource is at carrying capacity, and that all cells are in groups when the resource is completely depleted.”

10. In line 101, should “survive” be replaced by “reproduce”?

11. Lines 167-168: “Numerical simulations indicate ... it does not diverge”. While numerical simulations can be suggestive (although I am not so confident they are, in this case), they are not by themselves sufficient to claim a function diverges. This claim should be removed. I do think the authors’ comments on finite populations (170—173) are intuitive and helpful; these considerations also do not depend on the exact asymptotic behavior of the period of the oscillations.

12. Consider rewording lines 138-140. Saying that slow cells spread within aggregates can be misleading, because the fraction of slow cells that belong to aggregates is constant.

13. In line 180, “the time-scale separation is highest close to the onset of the oscillations.” I don’t understand how the authors get this from Figure 3. For example, if I take \\lambda_F=10 and increase \\lambda_S, the period of oscillations only increases.

14. Line 182: remain -> remains.

15. Line 210, what is meant by “the dominant Lyapunov exponent ‘transverse’ to the resident limit cycle”?

16. In line 216, “the emergence of a new collective time scale constrains its effects to a single phase of the life cycle”. Is this really true? Cheating is always constricted to the group phase of the life cycle by the way the model is constructed.

17. The suggestion in lines 219-229, that slow-down of evolutionary dynamics at high levels of cheating may be a mechanism for coexistence, I found quite enticing.

18. Mutations are required to transition between fast and slow cells. There may be interesting parallels with Paul Rainey’s work on the emergence of multicellular life cycles, because the life cycles in his work also initially depend on mutations (e.g. Hammerschmidt et al. 2014). The discussion in lines 280-282 could be expanded if the authors think these parallels make sense.

19. The discussion of parallels with D. discoideum (paragraph starting at line 284) belongs in the Introduction, because these observations serve as motivation for the model.

20. Line 300-303 add little beyond what is said in the Results section, and can be removed.

Reviewer #3: Already given in the attached file.

**Have all data underlying the figures and results presented in the manuscript been provided?**

Reviewer #1: Yes

Reviewer #2: Yes

Reviewer #3: Yes

PLOS authors have the option to publish the peer review history of their article (what does this mean?). If published, this will include your full peer review and any attached files.

Reviewer #1: No

Reviewer #2: No

Reviewer #3: No
---

## [Decision Letter · Decision Letter 1]

24 Nov 2020

Dear Mr. Miele,

Thank you very much for submitting your manuscript "Aggregative cycles evolve as a solution to conflicts in motility investment" for consideration at PLOS Computational Biology. As with all papers reviewed by the journal, your manuscript was reviewed by members of the editorial board and by several independent reviewers.

Both the reviewers and I appreciated the attention to an important topic and the thoughtful responses and changes following the first set of comments. Based on the reviews, we will accept this manuscript for publication. However, I wanted to give you an opportunity to consider the remaining few comments from the two reviewers and see if you decide to make any final changes in response. Acceptance is not contingent upon this and we do not request a response to the reviewers; the paper will not be sent back to review. But I do encourage you to consider the comments, especially the ones that Reviewer 3 reiterates.

Sincerely,

Corina E. Tarnita

Associate Editor

PLOS Computational Biology

Stefano Allesina

Deputy Editor

PLOS Computational Biology

[LINK]

Reviewer's Responses to Questions

**Comments to the Authors:**

Reviewer #2: I reread the revised version with interest and I believe the manuscript has substantially improved. I found that the presentation and the more conservative interpretation of the emergent cycles as "life-like cycles" rather than "life cycles" do more justice to the interesting results in this paper, which I (even more so when reading this paper again in its revised form) found fascinating and thought-provoking. In sum, the revised version was, at least for me, much more effective in driving home what makes the results of this paper novel and interesting.

I'm also happy with the authors' detailed and thoughtful responses to my comments, including those that challenged the authors' interpretation and those that arose from a misunderstanding on my end.

I only have a small number of remaining comments and suggestions.

1. In the abstract, and the discussion, the author write the emergent oscillations are an "adaptive response to [the escalation of conflicts]". I'm not sure what to make of this claim. Adaptive to whom?

2. Line 5: transition -> transitions.

3. Line 30 could perhaps be more compelling if this paragraph on previous work ends by explicitly stating what is missing from previous work.

4. Over all I found the explanation in the introduction very clear.

5. Line 101: context -> social context.

6. Line 124-125: I was thrown off by the added statement in the parentheses. I noted that this was in response to another reviewer's comments, but it's not clear what the "without loss of generality" statement applies to.

7. In the caption of Figure 2, mention what the dashed line is.

8. Line 196: is -> are.

9. Line 197: Here a reference to "life cycle" remains.

10. Line 216: Similarly, I think here the authors want to say "life-like cycle".

11. Line 330: cell -> cells.

Congratulations on this nice work! I look forward to seeing it in print.

Reviewer #3: Most of my comments have been addressed satisfactorily. Now I understand what the authors mean by "emergence"; it is the emergence of a life cycle, not emergence of multicellularity, which was my impression at first. Incidentally, unlike the other reviewer, I see no objection to using "life cycle". Indeed, "life-like cycle" sounds rather awkward. There are three points which I would like the authors to consider once again, but do not insist that they accept my advice.

(a) I agree that motility is an excellent candidate for the basis of what makes the two cell types different. Also, as the authors rightly say, it strengthens the comparison with the documented Dictyostelium example. But my point was that motility plays no role in the model. Ipso facto, motility differences play no role either. The model demonstrates how temporal oscillations in a well-mixed heterogeneous system, along with two time scales, can emerge as an automatic consequence of three factors: the dynamics of growth, resource level-dependent aggregative behaviour, and frequency-dependent fitness. That being so, the use of "motility" in the title (now amended to "Aggregative cycles evolve as a solution to conflicts in motility investment") is not correct. A more appropriate title should be chosen. The text can remain unchanged, except that a sentence can be added stating that motility is a plausible candidate for the differential phenotypic trait they are considering.

(b) The factor R/K appears naturally in the equation dR/dt = R[r(1-R/K) -N], because it shows that the intrinsic rate of growth of the resource is zero when the carrying capacity of the environment has been reached. However, it is a different matter to say that the same R/K is also the probability that a fast cell remains unaggregated, because that gives R/K an additional role in the model. It affects the payoffs, and therefore the dynamical consequences of equations (1). The authors say that the 1/K factor is used simply to make R dimensionless, and any other factor with the same dimension as 1/K would do. They may be right. But I am unable to see it immediately. Readers will benefit if the authors provide an explicit demonstration in the supplementary file. The text can just make a mention of it.

(c) The description of cell speed as a public good continues to bother me. The authors explain it as the benefit conferred by fast cells to slow cells that are in the same collective. It seems to me that in order to qualify as a public good, the same benefit must be available to other fast cells in the group too. It would amount to saying that a collective made up only of fast cells will move faster than any fast cell could by itself. That is plausible as a correlate of the size-dependence of speed, as found in the case of Dictyostelium slugs (Inouye Κ and Takeuchi I 1979 Analytical studies on migrating, movement of the pseudoplasmodium of Dictyostelium discoideum; Protoplasma 99: 289-304). The authors could make use of this finding to justify their assumption. But it would still not qualify for the designation of public good.

**Have all data underlying the figures and results presented in the manuscript been provided?**

Reviewer #2: None

Reviewer #3: Yes

PLOS authors have the option to publish the peer review history of their article (what does this mean?). If published, this will include your full peer review and any attached files.

Reviewer #2: No

Reviewer #3: No
---

## [Editor Report · Decision Letter 2]

7 Dec 2020

Dear Mr. Miele,

We are pleased to inform you that your manuscript 'Aggregative cycles evolve as a solution to conflicts in social investment' has been provisionally accepted for publication in PLOS Computational Biology.

Best regards,

Corina E. Tarnita

Associate Editor

PLOS Computational Biology

Stefano Allesina

Deputy Editor

PLOS Computational Biology

---

## [Editor Report · Acceptance letter]

15 Jan 2021

PCOMPBIOL-D-20-01189R2 

Aggregative cycles evolve as a solution to conflicts in social investment

Dear Dr Miele,

I am pleased to inform you that your manuscript has been formally accepted for publication in PLOS Computational Biology. Your manuscript is now with our production department and you will be notified of the publication date in due course.

With kind regards,

Jutka Oroszlan
